# Glucocorticoid receptor-mediated amygdalar metaplasticity underlies adaptive modulation of fear memory by stress

Ran Inoue[1]*, Kareem Abdou[2,3], Ayumi Hayashi-Tanaka[1], Shin-ichi Muramatsu[4,5], Kaori Mino[1], Kaoru Inokuchi[2], Hisashi Mori[1]

[1]Department of Molecular Neuroscience, Graduate School of Medicine and Pharmaceutical Sciences, University of Toyama, Toyama, Japan; [2]Department of Biochemistry, Graduate School of Medicine and Pharmaceutical Sciences, University of Toyama, Toyama, Japan; [3]Department of Biochemistry, Faculty of Pharmacy, Cairo University, Cairo, Egypt; [4]Division of Neurology, Department of Medicine, Jichi Medical University, Tochigi, Japan; [5]Center for Gene and Cell Therapy, The Institute of Medical Science, The University of Tokyo, Tokyo, Japan

**Abstract** Glucocorticoid receptor (GR) is crucial for signaling mediated by stress-induced high levels of glucocorticoids. The lateral nucleus of the amygdala (LA) is a key structure underlying auditory-cued fear conditioning. Here, we demonstrate that genetic disruption of GR in the LA (LAGRKO) resulted in an auditory-cued fear memory deficit for strengthened conditioning. Furthermore, the suppressive effect of a single restraint stress (RS) prior to conditioning on auditory-cued fear memory in floxed GR (control) mice was abolished in LAGRKO mice. Optogenetic induction of long-term depression (LTD) at auditory inputs to the LA reduced auditory-cued fear memory in RS-exposed LAGRKO mice, and in contrast, optogenetic induction of long-term potentiation (LTP) increased auditory-cued fear memory in RS-exposed floxed GR mice. These findings suggest that prior stress suppresses fear conditioning-induced LTP at auditory inputs to the LA in a GR-dependent manner, thereby protecting animals from encoding excessive cued fear memory under stress conditions.
DOI: https://doi.org/10.7554/eLife.34135.001

*For correspondence: eiran@med.u-toyama.ac.jp

## Introduction

Stress activates the hypothalamus–pituitary–adrenal (HPA) axis, which results in the release of glucocorticoid hormones (cortisol in humans and corticosterone in rodents) from the adrenal cortex. Glucocorticoid hormones can readily enter the brain and bind to specific receptors in regions crucial for memories of stressful experiences, such as the hippocampus and amygdala, thereby enhancing the consolidation of emotionally arousing events (*de Quervain et al., 2009*; *Roozendaal et al., 2009*). Two types of receptors mediate the effects of glucocorticoids: type I mineralocorticoid receptor (MR, *Nr3c2*) and type II glucocorticoid receptor (GR, *Nr3c1*). Compared with MR, GR has a lower binding affinity for glucocorticoids and is largely unoccupied at basal levels, and is thus considered to be particularly important in signaling mediated by stress-induced high levels of glucocorticoids (*de Kloet et al., 2005*; *Reul and de Kloet, 1985*).

The lateral nucleus of the amygdala (LA) is a key structure underlying auditory-cued fear conditioning (AFC) (*LeDoux, 2000*). Auditory information, which is critical for AFC, reaches the LA either from the medial geniculate nucleus (MGN) or from the auditory cortex (AC). Long-term potentiation

(LTP) in the two auditory inputs to the LA is essential for the acquisition and expression of auditory-cued fear memory (*Blair et al., 2001*; *Rogan et al., 1997*; *Tsvetkov et al., 2002*). In response to stress, GR activation plays a central role in the formation of long-term memory, which is an essential mechanism for learning from stressful events and respond adaptively to similar demands in the future (*Finsterwald and Alberini, 2014*). In an electrophysiological study using brain slices, glucocorticoid prolonged excitatory synaptic responses in the basolateral complex of the amygdala (BLA) by binding to GR (*Karst et al., 2010*). In contrast, after an acute stress exposure, application of glucocorticoid suppressed excitatory synaptic responses in the BLA in a GR-dependent manner. This switch in synaptic response to glucocorticoid is referred to as metaplasticity (*Abraham and Tate, 1997*; *Schmidt et al., 2013*). Prior delivery of behavioral stress has also been shown to suppress subsequent induction of LTP (*Kavushansky and Richter-Levin, 2006*). However, whether stress-induced amygdalar metaplasticity occurs in vivo in a way relevant to the strength of the auditory-cued fear memory, and what role LAGR plays in this neural process are unknown.

In this study, we generated LA-selective GR knockout (LAGRKO) mice to investigate the region-specific role of LAGR in mediating the modulatory effects of stress on fear memory. We first compared contextual and auditory-cued fear memory conditioned at different strengths between floxed GR (control) and LAGRKO mice at basal condition. We then investigated the effect of LAGR disruption on the adaptive modulation of fear memory after exposure to acute stress. Finally, using an optogenetic technique to induce LTP and long-term depression (LTD), we investigated how GR-dependent metaplasticity in the LA influenced auditory-cued fear memory in response to prior stress exposure.

## Results

### Generation of LAGRKO mice

To selectively knockout the GR gene in the LA, we generated two mutant mouse lines. The knock-in line expressed improved Cre (iCre) recombinase (*Shimshek et al., 2002*) under the control of the gastrin-releasing peptide gene (*Grp*) promoter, and the animals are referred to as Grp-iCre mice (*Figure 1A,B*). Grp is abundant in the LA and absent or present at low levels in other subnuclei of the amygdala (*Shumyatsky et al., 2002*). Thus, *Grp* promoter could be used for driving the expression of Cre recombinase selectively in the LA. To confirm its usefulness, we crossed a Grp-iCre mouse with a CAG-CAT-Z reporter mouse, which carried the chloramphenicol acetyltransferase gene (*CAT*) flanked by two loxP sites and the β-galactosidase gene in sequence (*Araki et al., 1995*). Cre-mediated recombination between the two loxP sites resulted in the expression of β-galactosidase, which was detected by X-gal staining. As shown in *Figure 1C*, strong β-galactosidase expression was detected in the LA and hippocampal CA3 region of Grp-iCre/CAG-CAT-Z mice, indicating the presence of robust Cre-loxP recombination in these brain regions. In addition, sparse and weak expression of β-galactosidase appeared in the accessory basal nucleus of the amygdala and in layer 6 of the cerebral cortex.

Next, we generated a floxed GR ($Nr3c1^{loxP/loxP}$) mouse line in which exon 3 of the $Nr3c1$ gene (encoding the DNA-binding domain) was flanked by two loxP sites (*Figure 1—figure supplement 1*). Grp-iCre mice were crossed with floxed GR mice to establish the LAGRKO mouse line ($Nr3c1^{loxP/loxP}$, $Grp^{iCre+/−}$). Immunofluorescence staining with a specific antibody against GR revealed a selective disruption of GR proteins in the LA of LAGRKO mice (*Figure 2A*). There was no significant difference between the two genotypes in the expression level of GR in the central nucleus and basal nucleus of the amygdala. Double immunofluorescence staining with anti-GR and anti-NeuN (a neuronal marker) antibodies showed that GR was undetectable in approximately 70% of LA neurons in LAGRKO mice (floxed GR, 90.88 ± 1.39%; LAGRKO, 20.27 ± 0.95%; *Figure 2A,B*, *Figure 2—figure supplement 1A*). The expression level of GR in the hippocampal CA3 region of floxed GR mice was very low, and it was slightly decreased in LAGRKO mice (*Figure 2C*). There was no significant difference between the two genotypes in the expression level of GR in the cerebral cortex (*Figure 2C*, *Figure 2—figure supplement 1B*), hippocampal CA1 and CA2 regions, and dentate gyrus (*Figure 2C*). Collectively, these results indicate the successful establishment of a novel LAGRKO mouse line.

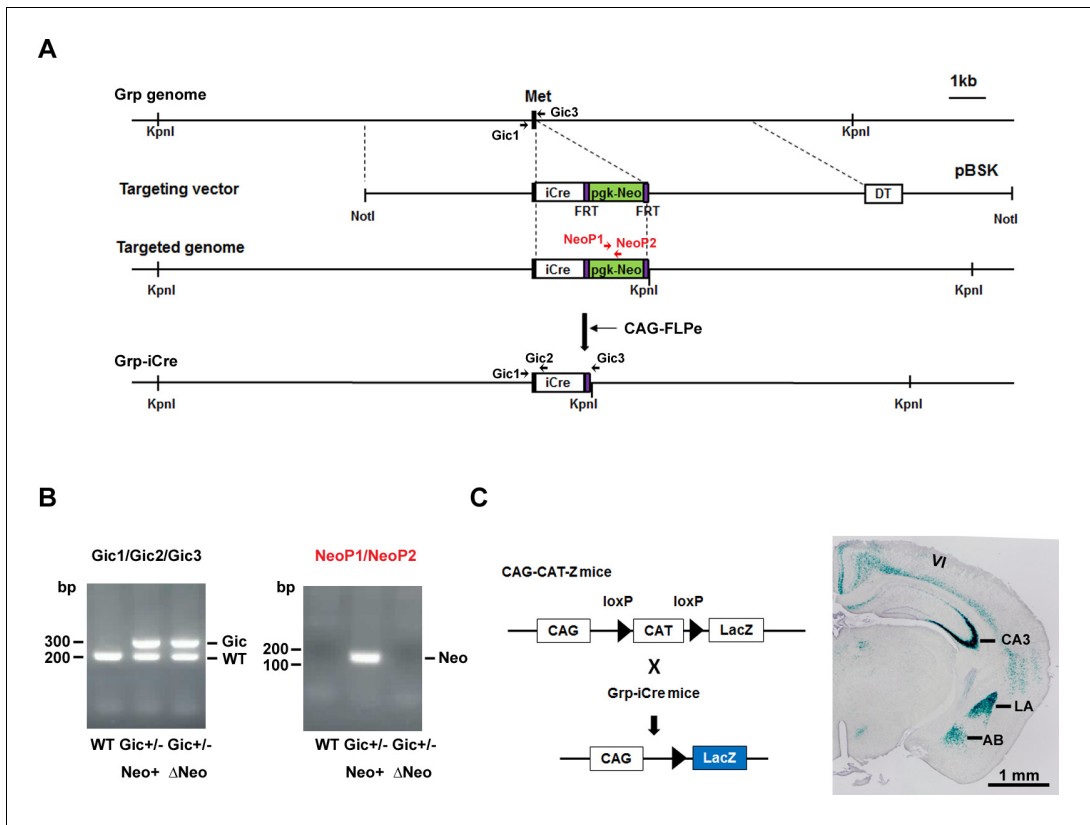

**Figure 1.** Generation and characterization of Grp-iCre mice. (**A**) Schematic diagram of the gene targeting strategy. iCre and pgk-neo cassettes flanked by two FRT sites were inserted into the gastrin-releasing peptide gene (*Grp*) locus. Met is the translation initiation site of Grp. The location of PCR primers (Gic1, Gic2, Gic3, NeoP1, and NeoP2) used for genotyping are indicated. DT, diphtheria toxin gene; pBSK, pBluescriptII SK. The chimeric mouse obtained was crossed with a CAG-FLPe mouse to delete the pgk-neo cassette and establish the Grp-iCre (Gic) mouse line. (**B**) Genotyping PCR of genomic DNA prepared from WT; $Gic^{+/-}$, $Neo^+$; $Gic^{+/-}$, and ΔNeo mice. (**C**) Cre activity in Grp-iCre mice was examined by crossing Grp-iCre mice with lacZ reporter mice (CAG-CAT-Z) mice. β-galactosidase expression in a Grp-iCre/CAG-CAT-Z mouse brain stained with X-gal. X-gal staining revealed robust Cre-loxP recombination in the lateral nucleus of the amygdala (LA) and the hippocampal CA3 region, with sparser recombination in the accessary basal nucleus of the amygdala (AB), and in layer 6 of the cerebral cortex (VI).

DOI: https://doi.org/10.7554/eLife.34135.002

The following figure supplement is available for figure 1:

**Figure supplement 1.** Generation of floxed GR mice.

DOI: https://doi.org/10.7554/eLife.34135.003

## Disruption of LAGR impairs auditory-cued fear memory for strengthened conditioning

Since GR has a low binding affinity for glucocorticoids and its activation requires high levels of hormones, we investigated the role of LAGR in AFC at different strengths. After moderate conditioning, which included three CS ×US pairings, LAGRKO mice exhibited the same freezing levels as floxed GR mice during the training, contextual test, and auditory-cued test (*Figure 3—figure supplement 1*). When the training was strengthened to six CS × US pairings (*Figure 3A*), LAGRKO mice exhibited significantly lower freezing levels than floxed GR mice during tone presentation in the cued test (p=0.046; unpaired Student's *t* test; *Figure 3D*), whereas there were no significant differences between the two genotypes in freezing levels during the training ($F_{1,126}$ = 1.19, p=0.278; two-way repeated measures ANOVA; *Figure 3B*) and contextual test (p=0.803; unpaired Student's *t* test; *Figure 3C*). Corticosterone assays revealed significantly higher plasma corticosterone levels 90 min after training with six CS × US pairings than after training with three CS × US pairings in both

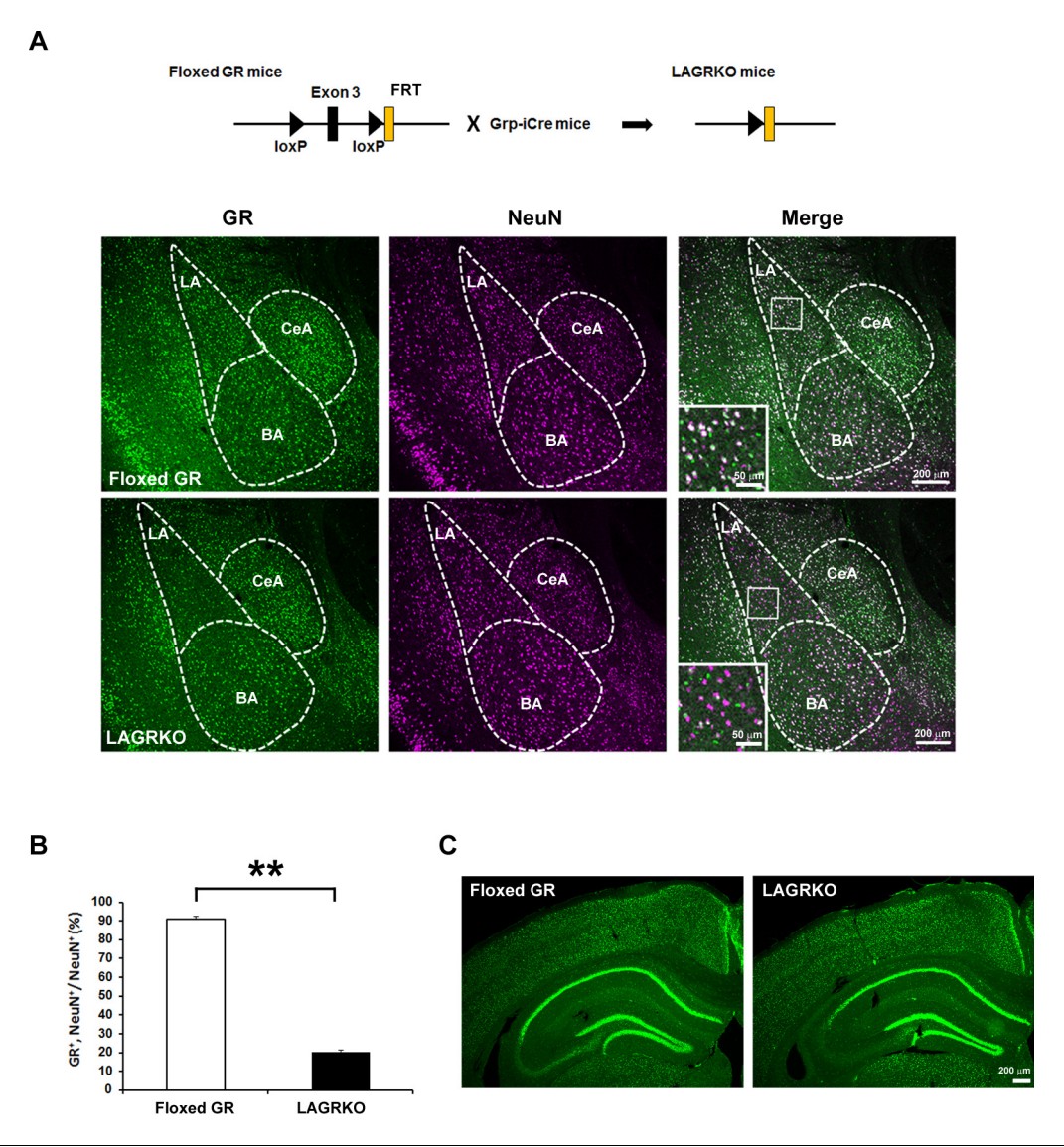

**Figure 2.** Generation and characterization of lateral amygdala (LA)-selective glucocorticoid receptor (GR) knockout (LAGRKO) mice. (**A**) The LAGRKO mouse line (GR$^{flox/flox}$, Grp-iCre+/-) was established by crossing floxed GR (GR$^{flox/flox}$) and Grp-iCre mice. Double immunofluorescence staining of GR (green, left panels) and NeuN (magenta, middle panels) in coronal brain sections from floxed GR and LAGRKO mice. The overlap of green and magenta signals (white, right panels) indicates the expression of GR in LA neurons in floxed GR mice (upper), which was apparently reduced in LAGRKO mice (lower). Magnified images of the boxed areas are shown in the insets. LA, lateral nucleus of the amygdala; BA, basal nucleus of the amygdala; CeA, central nucleus of the amygdala. (**B**) Quantification of GR$^+$ and NeuN$^+$ cells in the LA of floxed GR and LAGRKO mice (n = 9 sections from three mice). Data are presented as mean ± S.E.M. **p<0.001. (**C**) Expression of GR in the cerebral cortex and hippocampus.

DOI: https://doi.org/10.7554/eLife.34135.004

The following source data and figure supplement are available for figure 2:

**Source data 1.** Raw data for generating *Figure 2B*.
DOI: https://doi.org/10.7554/eLife.34135.006
**Figure supplement 1.** Expression of GR in the LA and cerebral cortex of floxed GR and LAGRKO mice.
DOI: https://doi.org/10.7554/eLife.34135.005

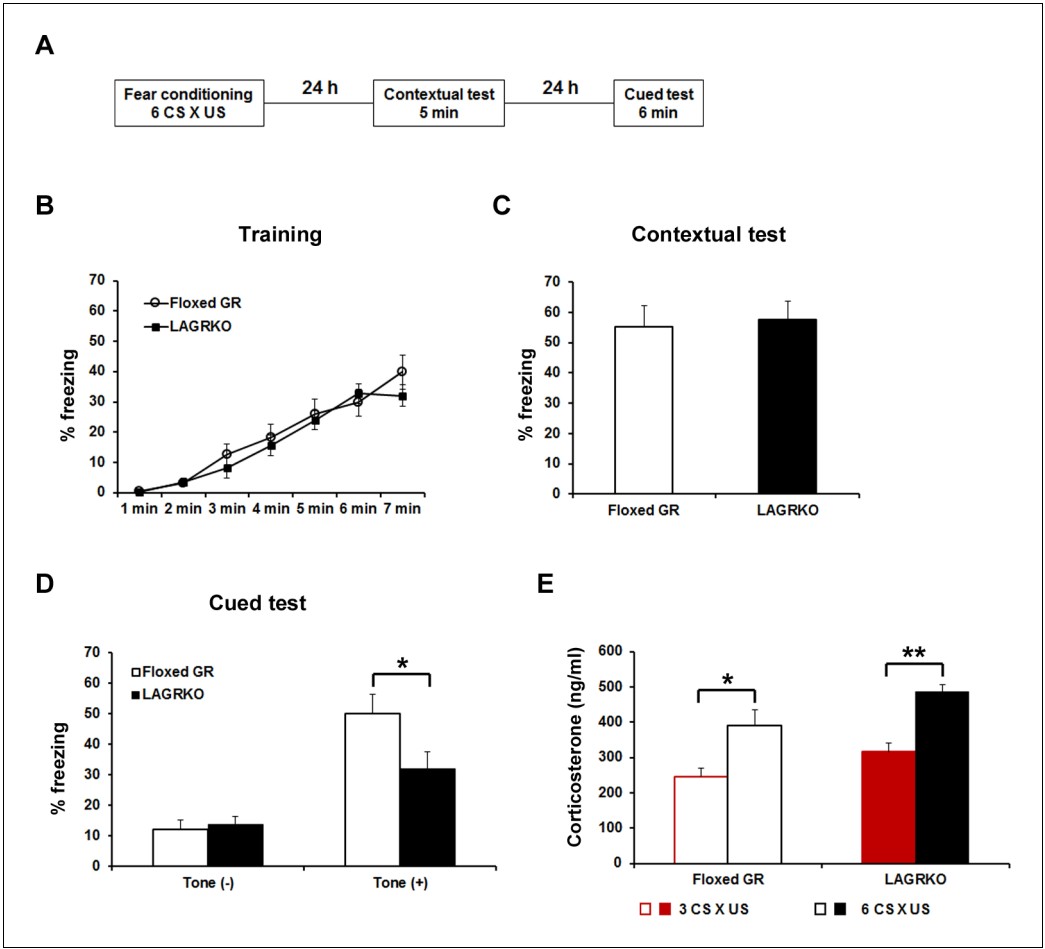

**Figure 3.** Genetic disruption of LAGR impairs auditory-cued fear memory following strengthened conditioning. (**A**) Experimental protocol for fear conditioning. Mice were trained with a protocol using six CS × US pairings. Contextual and auditory-cued fear memory were tested 24 and 48 hr after training, respectively. Freezing levels during the training (**B**), contextual test (**C**), and cued test (**D**). During the cued test, LAGRKO mice (n = 10) exhibited significantly lower freezing levels in the presence of the tone than did floxed GR mice (n = 10). There was no significant difference in freezing levels between the two genotypes during the training and contextual test. (**E**) Plasma corticosterone levels 90 min after training were significantly higher in mice conditioned with six CS × US pairings (n = 7) than in mice conditioned with 3 CS × US pairings (n = 6) in both genotypes. Data are presented as mean ± S. E. M. *p<0.05, **p<0.001.

DOI: https://doi.org/10.7554/eLife.34135.007

The following source data and figure supplements are available for figure 3:

**Source data 1.** Raw data for generating *Figure 3B–E*.
DOI: https://doi.org/10.7554/eLife.34135.010
**Figure supplement 1.** LAGRKO mice did not exhibit fear memory deficits with the moderate fear conditioning protocol.
DOI: https://doi.org/10.7554/eLife.34135.008
**Figure supplement 2.** LAGRKO mice exhibited no changes in locomotor activity and anxiety level as determined using the open field test.
DOI: https://doi.org/10.7554/eLife.34135.009

genotypes (floxed GR, p=0.016; LAGRKO, p<0.001; unpaired Student's *t* test; *Figure 3E*). In the open field test, LAGRKO mice did not show any alteration in anxiety level or locomotion (*Figure 3—figure supplement 2*), suggesting that the deficit of auditory-cued fear memory in LAGRKO mice was not the result of altered anxiety and locomotor activity.

## Injection of GR-expressing adeno-associated virus into the LA rescues the auditory-cued fear memory deficit in LAGRKO mice

Although we successfully and selectively disrupted GR in the LA of LAGRKO mice, we cannot exclude the possibility that sparse disruption of GR in other brain areas, such as the hippocampal CA3 region, may be associated with behavioral impairment in LAGRKO mice. To confirm that the auditory-cued fear memory deficit can be specifically attributed to the disruption of LAGR, we performed a rescue experiment by injecting a GR-expressing adeno-associated virus (AAV-GR) into the LA of LAGRKO mice. Green fluorescent protein-expressing AAV (AAV-GFP) was used as the control. Fear conditioning was conducted 4 weeks after virus injection (*Figure 4A,B*). LAGRKO mice injected with AAV-GR and floxed GR mice injected with AAV-GFP exhibited significantly higher freezing

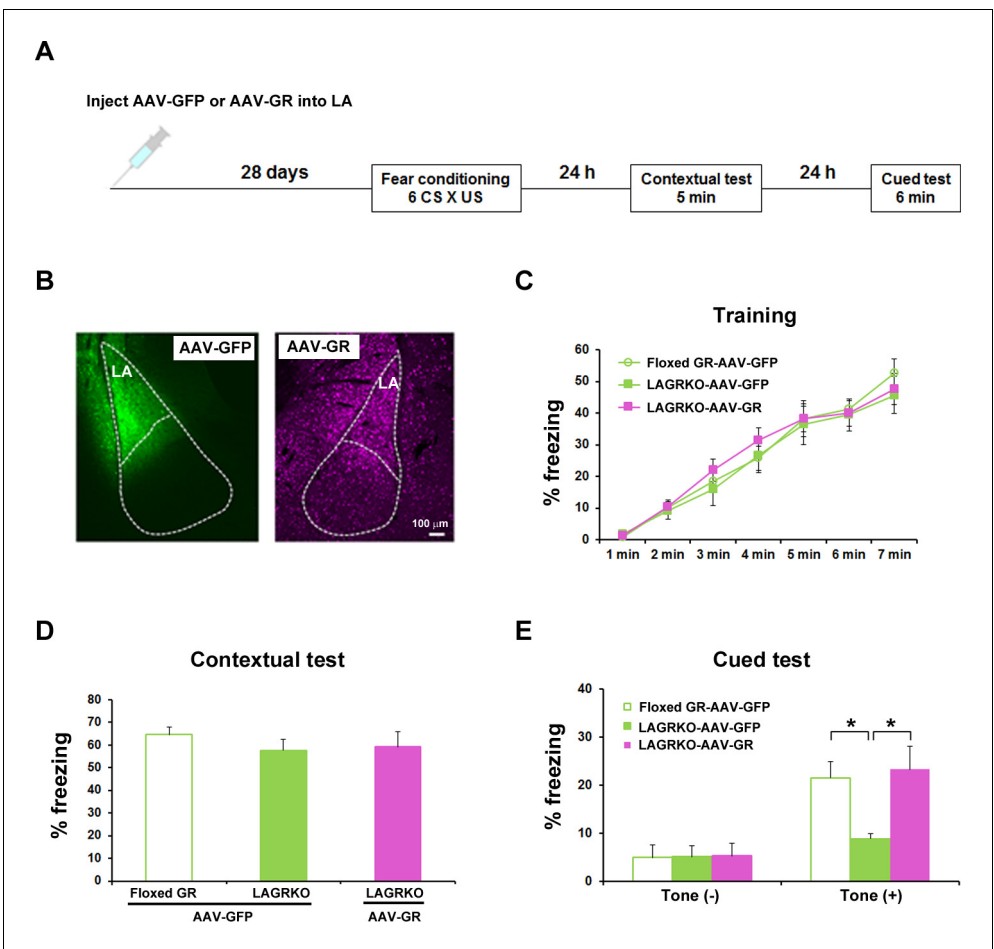

**Figure 4.** Rescue of the auditory-cued fear memory deficit in LAGRKO mice by restoring GR expression levels in the LA. (**A**) The experimental protocol for adeno-associated virus (AAV) vector injection and fear conditioning. (**B**) Representative images of the expression of green fluorescent protein (GFP, green) and GR (magenta) in the LA of LAGRKO mice injected with AAV-GFP or AAV-GR. The expression levels of GFP and GR were assessed after the behavioral test was completed. Freezing levels during the training (**C**), contextual test (**D**), and cued test (**E**). During the cued test, LAGRKO mice (n = 9) injected with AAV-GR and floxed GR injected with AAV-GFP exhibited significantly higher levels of freezing in the presence of the tone than did AAV-GFP-injected LAGRKO mice (n = 9). There were no significant differences between the three groups during the training and contextual tests. Data are presented as mean ± S. E. M. *p<0.05.

DOI: https://doi.org/10.7554/eLife.34135.011

The following source data is available for figure 4:

**Source data 1.** Raw data for generating *Figure 4B–E*.

DOI: https://doi.org/10.7554/eLife.34135.012

levels during the auditory-cued fear memory test than did LAGRKO mice injected with AAV-GFP ($F_{2,24}$ = 5.25, p=0.013; one-way ANOVA followed by *post hoc* Tukey–Kramer test; *Figure 4E*). There was no significant difference in freezing levels between the three mouse groups during training ($F_{2,168}$ = 0.60, p=0.522; two-way repeated measures ANOVA; *Figure 4C*) and the contextual fear memory test ($F_{2,24}$ = 0.53, p=0.600; one-way ANOVA; *Figure 4D*). This rescue experiment further suggests the specific involvement of LAGR in AFC.

### Disruption of LAGR impairs adaptive modulation of conditioned fear in response to prior restraint stress exposure

Fear conditioning is highly susceptible to modulation by prior stress exposure (*Cordero et al., 2003*; *Rodríguez Manzanares et al., 2005*). To investigate the involvement of LAGR in mediating the effect of prior stress on subsequent fear conditioning, floxed GR and LAGRKO mice were exposed to a 20 min restraint stress (RS) and conditioned 1 hr later (*Figure 5A*). Mice were conditioned with three CS × US pairings that resulted in equivalent freezing levels in floxed GR and LAGRKO mice in the contextual and auditory-cued fear memory tests (*Figure 3—figure supplement 1*). RS exposure 1 hr prior to fear conditioning induced a significant increase in freezing levels during the training ($F_{3,215}$ = 32.25, p<0.001; two-way repeated measures ANOVA; *Figure 5B*) and contextual memory test ($F_{3,43}$ = 5.03, p=0.004; one-way ANOVA followed by *post hoc* Tukey–Kramer test; *Figure 5C*) in both floxed GR and LAGRKO mice. In contrast to the facilitating effect of prior RS exposure on contextual fear memory, RS exposure 1 hr before fear conditioning induced a significant decrease in auditory-cued freezing in floxed GR mice but not in LAGRKO mice (floxed GR, p=0.016; LAGRKO, p=0.951; unpaired Student's *t* test; *Figure 5D*), suggesting this suppressive effect is LAGR-dependent. In order to examine whether RS and AFC induce differences in the levels of circulating corticosterone, plasma corticosterone levels were measured 1 hr after RS exposure alone and 90 min after RS exposure followed by AFC. There was no significant difference in plasma corticosterone levels observed between floxed GR and LAGRKO mice at the two time points (1 hr after RS exposure alone, p=0.21; 90 min after RS exposure followed by AFC, p=0.67; unpaired Student's *t* test; *Figure 5—figure supplement 1*).

### Optogenetic manipulation of metaplasticity in the LA alters the behavioral effect of prior RS exposure on auditory-cued fear memory

We next investigated whether stress-induced changes in synaptic plasticity were responsible for the effect of prior stress on auditory-cued fear memory in floxed GR and LAGRKO mice. It was recently proposed that metaplasticity may play a role in the regulation of learning and memory under stress (*Joëls et al., 2012*; *Myers et al., 2014*). Therefore, we hypothesized that prior RS exposure would suppress AFC-induced LTP in the LA, leading to the attenuation of auditory-cued fear memory in floxed GR mice, and that LAGR disruption might impair stress-induced metaplasticity in the LA. To test this, we used a recently established technique that allows engineering a memory using an optical LTP or LTD protocol (*Nabavi et al., 2014*). We injected an AAV vector expressing a variant of light-activated channelrhodopsin 2, oChIEF, into bilateral MGN and AC. Four weeks after virus injection, when oChIEF had reached axon terminals in the LA, mice were exposed to a RS and conditioned 1 hr later (*Figure 6A,B*). Immediately after the first auditory-cued fear memory test (Test 1), which was conducted 24 hr after fear conditioning, mice received an optical LTP or LTD protocol and tested 24 hr later (Test 2). The optical LTP protocol significantly increased freezing levels during the Test 2 compared with Test 1 in floxed GR mice but not in LAGRKO mice (floxed GR, p=0.046; LAGRKO, p=0.811; paired Student's *t* test; *Figure 6C,D*). These results support our hypothesis that prior RS exposure suppresses AFC-induced LTP in a GR-dependent manner, thus resulting in reduced auditory-cued fear memory in floxed GR mice. In LAGRKO mice, the optical LTD protocol significantly reduced freezing levels during Test 2 compared with that during Test 1 (p=0.010; paired Student's *t* test; *Figure 6E*), further supporting the hypothesis that prior stress weakens synaptic strength in the LA leading to reduced cued fear memory.

     To confirm that the optical LTP or LTD protocols produced the expected synaptic effects, we conducted in vivo recordings in the LA of anaesthetized naïve mice expressing oChIEF in auditory regions. Brief light pulses (0.033 Hz) at the recording site produced field responses, which were potentiated by the LTP protocol (Floxed GR, $F_{1,126}$ = 191.5, p<0.001; LAGRKO, $F_{1,126}$ = 237.7,

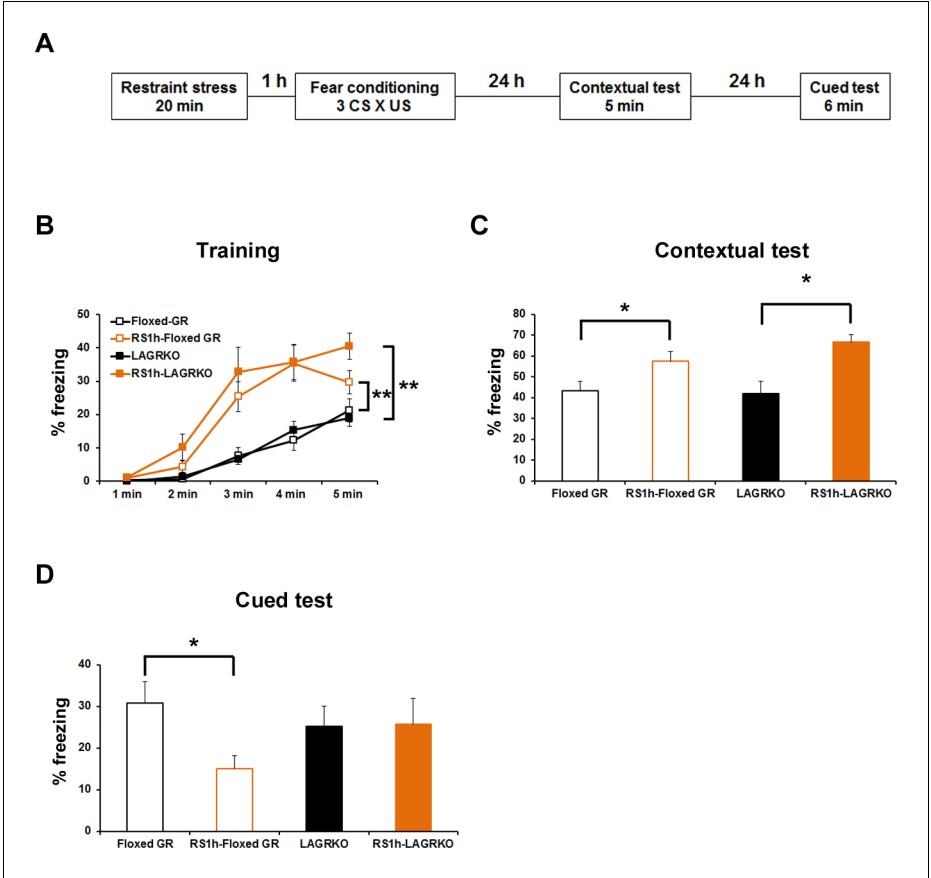

**Figure 5.** Effect of acute prior stress on auditory-cued fear conditioning. (**A**) The experimental design for prior restraint stress (RS) exposure and fear conditioning. Floxed GR and LAGRKO mice were exposed to a 20 min RS and fear-conditioned 1 hr later. Contextual and auditory-cued fear memories were tested 24 and 48 hr after training, respectively. (**B**) Mice exposed to RS 1 hr before conditioning (brown, RS1h-Floxed GR, n = 11; RS1h-LAGRKO, n = 10) exhibited significantly higher freezing levels than did nonstressed mice (black, Floxed GR, n = 10; LAGRKO, n = 16) during the training session in both genotypes. (**C**) Mice exposed to RS exhibited significantly higher freezing levels than did nonstressed mice during the contextual fear memory test. (**D**) During the cued test, floxed GR mice exposed to RS 1 hr before training exhibited significantly lower freezing levels than did the nonstressed floxed GR mice. Previous RS exposure had no effect on auditory-cued fear memory in LAGRKO mice. Data are presented as mean ± S. E. M. *p<0.05; **p<0.001.
DOI: https://doi.org/10.7554/eLife.34135.013

The following source data and figure supplement are available for figure 5:

**Source data 1.** Raw data for generating *Figure 5B–D*.
DOI: https://doi.org/10.7554/eLife.34135.015
**Figure supplement 1.** Restraint stress- and auditory fear conditioning-induced release of corticosterone was comparable between floxed GR and LAGRKO mice.
DOI: https://doi.org/10.7554/eLife.34135.014

p<0.001; two-way repeated measures ANOVA; *Figure 6F*) and depressed by the LTD protocol (Floxed GR, $F_{1,126} = 511.9$, p<0.001; LAGRKO, $F_{1,126} = 497.3$, p<0.001; two-way repeated measures ANOVA; *Figure 6G*) in both floxed GR and LAGRKO mice. These results suggest that the LTP and LTD protocols used in the behavioral experiments modify synaptic strength in the expected manner.

## Discussion

In the present study, we demonstrate that LAGR exclusively mediates auditory-cued fear responses to strengthened conditioning, suggesting a critical role for LAGR in signaling mediated by stress-

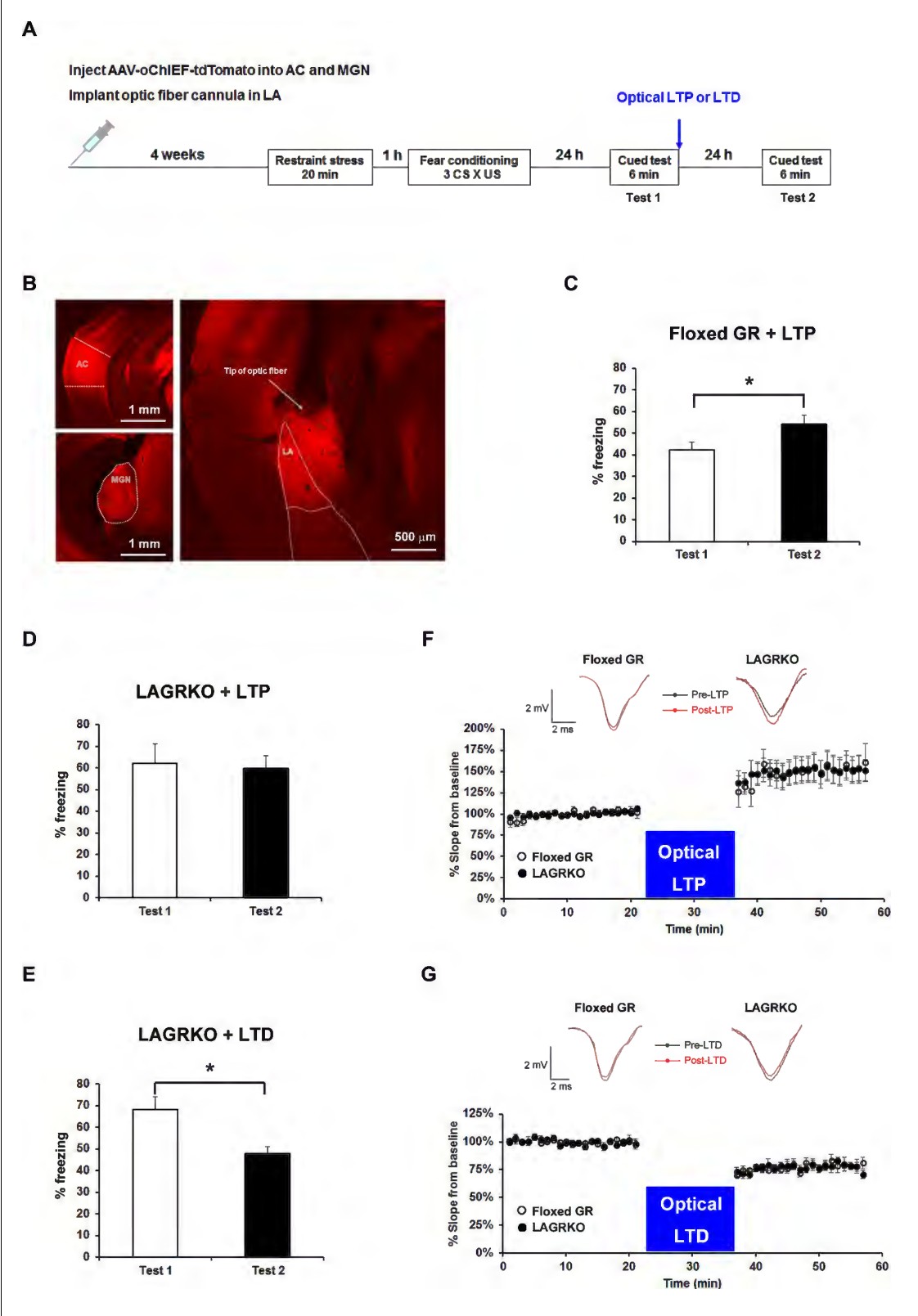

**Figure 6.** Optogenetic manipulation of LTP or LTD at auditory inputs to the LA alters the effect of acute stress on auditory-cued fear memory. (**A**) Experimental schedule for AAV injection, RS exposure, auditory-cued fear conditioning, and optical LTP or LTD induction. (**B**) Representative fluorescence image of areas expressing AAV-oChIEF-tdTomato 4 weeks after virus injection in the medial geniculate nucleus (MGN) and auditory cortex (AC). LA, lateral nucleus of the amygdala. (**C**) Floxed GR mice (n = 8) that received immediate optical LTP after the first auditory-cued fear memory test
*Figure 6 continued on next page*

*Figure 6 continued*

(Test 1) exhibited significantly increased freezing levels in the second auditory-cued fear memory test (Test 2). (D) LAGRKO mice (n = 8) that received optical LTP stimulation exhibited the same freezing levels in Tests 1 and 2. (E) LAGRKO mice (n = 8) that received LTD stimulation immediately after Test 1 exhibited significantly lower freezing levels in Test 2 than in Test 1. Graphs show the freezing rate during the first 1 min of tone presentation at Tests 1 and 2. (F, G) A plot of the average of field EPSP slopes (normalized to the period before optical stimulation) before and after delivering an optical LTP (Floxed GR, n = 4, before LTP, 100 ± 0.8%, after LTP, 148.9 ± 3.2%; LAGRKO, n = 4, before LTP, 100 ± 0.6%, after LTP, 149.3 ± 2.8%) or LTD (Floxed GR, n = 4, before LTD, 100 ± 0.5%, after LTD, 77.4 ± 0.8%; LAGRKO, n = 4, before LTD, 100 ± 0.6%, after LTD, 76.7 ± 0.8%). Data are presented as mean ± S. E. M. *p<0.05.

DOI: https://doi.org/10.7554/eLife.34135.016

The following source data and figure supplement are available for figure 6:

**Source data 1.** Raw data for generating *Figure 6C–E*.
DOI: https://doi.org/10.7554/eLife.34135.018
**Source data 2.** Raw data for generating *Figure 6F,G*.
DOI: https://doi.org/10.7554/eLife.34135.019
**Figure supplement 1.** A putative model of LAGR-dependent mechanisms underlying alterations in synaptic plasticity during fear conditioning in basal condition or 1 hr after RS exposures.
DOI: https://doi.org/10.7554/eLife.34135.017

induced high levels of glucocorticoids. We further demonstrate that LAGR is indispensable for the suppressive effect of prior RS exposure on AFC. The optogenetic potentiation of synaptic strength at auditory inputs to the LA by an LTP protocol significantly increased auditory-cued fear memory in RS-exposed floxed GR but not in RS-exposed LAGRKO mice. Accordingly, weakening the synaptic strength of auditory inputs to the LA by an LTD protocol significantly reduced auditory-cued fear memory in RS-exposed LAGRKO mice. These findings suggest that stress-induced metaplasticity occurs in the LA in a GR-dependent manner, and is responsible for the suppressive effect of prior RS on AFC.

The degree of stress induced by learning itself has been shown to be related to strength of the memory (*Cordero et al., 1998*; *Cordero and Sandi, 1998*). In our study, there was no difference in freezing levels between floxed GR and LAGRKO mice conditioned with three CS × US pairings. When the number of CS × US pairings of the conditioning procedure was increased to six, LAGRKO exhibited an impaired auditory-cued fear memory. Concomitantly, in both genotypes, plasma corticosterone levels 90 min after fear conditioning were significantly higher in the group with six CS × US pairings than in the group with three CS × US pairings. It is likely that the increased number of CS × US pairings strengthened the connection between auditory CS and US in the LA, and LAGR, which is activated by high levels of circulating corticosterone, contributes exclusively to this neural process. These findings are in agreement with the lower binding affinity of GR for glucocorticoids, leading to the critical role of GR in signaling mediated by stress-induced high levels of glucocorticoids (*Reul and de Kloet, 1985*).

Emerging evidence indicates that stress exposure modulates subsequent learning and memory bidirectionally, either through facilitation or impairment (*Raio and Phelps, 2015*). Glucocorticoid signaling has been thought as a common mechanism for mediating the effects of stress on fear memory (*Cordero et al., 2003*; *Rodríguez Manzanares et al., 2005*). In our behavioral tests, a single RS exposure 1 hr before conditioning significantly reduced auditory-cued fear memory in floxed GR mice but not in LAGRKO mice, suggesting that the suppressive effect of prior RS is LAGR-dependent. To date, there are no reports demonstrating the effects of stress 1 hr before conditioning on fear memory. One previous report demonstrated that a single acute stress exposure 48 hr before fear conditioning had no effect on AFC (*Cordero et al., 2003*). GRs control the magnitude and duration of stress responses, and they are critical in the negative feedback regulation of the HPA axis in response to acute stress (*Laryea et al., 2015*). In our study, plasma corticosterone levels were significantly higher 1 hr after RS than 24 hr after RS (data not shown), and it is possible that the timing of previous stress exposure is a key determinant of the effects of stress on auditory-cued fear memory. Consistent with our in vivo evidence showing the LAGR-dependent suppressive effect of prior RS, Karst *et al.* demonstrated that a long-lasting glutamatergic transmission in BLA neurons triggered by the application of corticosterone was suppressed by RS 1 hr before slice preparation and the suppressive effect of RS was attenuated by treatment with a GR antagonist (*Karst et al., 2010*).

Synaptic changes at auditory inputs to the LA play an essential role in the acquisition and expression of auditory-cued fear memory (*Blair et al., 2001*; *Rogan et al., 1997*; *Tsvetkov et al., 2002*). Exposure to a single stress induces LTP in the BLA (*Sarabdjitsingh et al., 2012*; *Vouimba et al., 2004*). Therefore, it is possible that prior RS exposure occludes or suppresses subsequent AFC-induced LTP in floxed GR mice, thereby preventing the formation of excessive cued fear memory. To test this hypothesis, we induced LTP at auditory inputs to the LA by optogenetic stimulation, and found a significant increase in auditory-cued fear responses in RS-exposed floxed GR mice. This is the first evidence supporting a causal link between stress-induced metaplasticity in the LA and adaptive fear response to a salient cue in a stressful condition. Several in vitro and in vivo electrophysiological studies have demonstrated that the activation of GRs is involved in stress-induced amygdalar metaplasticity (*Karst et al., 2010*; *Kavushansky and Richter-Levin, 2006*). In our study, optogenetic delivery of an LTP protocol failed to alter fear responses to auditory cues in RS-exposed LAGRKO mice, supporting that the activation of LAGR is essential for stress-induced suppression of synaptic plasticity in the LA. The link between synaptic strength and fear response was further verified by results demonstrating that the weakening of the synaptic strength by delivering an optical LTD protocol attenuated auditory-cued fear memory in RS-exposed LAGRKO mice. In agreement with this finding, a previous electrophysiological study using brain slices containing the BLA region demonstrated that the persistence of LTP induced in the presence of corticosterone is sensitive to previous RS exposure (*Sarabdjitsingh and Joëls, 2014*). Herein, we did not directly examine the effects of a prior RS exposure on the subsequent induction or expression of LTP/LTD in fear-conditioned mice using an in vitro or in vivo electrophysiological method. However, the results of our optogenetic experiment, together with existing evidence (*Karst et al., 2010*; *Sarabdjitsingh and Joëls, 2014*), led us to postulate that the persistence of AFC-induced LTP is suppressed by a previous RS exposure, thereby protecting animals from encoding excessive cued fear memory.

Two previous studies have contributed to our understanding the GR-dependent mechanisms involved in prior stress-modulated synaptic plasticity. Di et al. demonstrated that a short application of corticosterone activates G protein-coupled postsynaptic membrane glucocorticoid receptors (mGRs), leading to the synthesis and release of an endocannabinoid, which binds to the presynaptic CB1 endocannabinoid receptor to suppress glutamate release (*Di et al., 2003*). Thus, postsynaptic mGRs activation induced by a recent RS exposure may result in reduction of glutamate release during subsequent fear conditioning. In another study, Sato et al. demonstrated that membrane-impermeable corticosterone treatment decreases $N$-methyl-D-aspartate (NMDA)-induced $Ca^{2+}$ elevation, suggesting that activation of mGRs may inhibit $Ca^{2+}$ influx via the NMDA receptor (*Sato et al., 2004*). Therefore, the reduced presynaptic glutamate release and inhibition of postsynaptic $Ca^{2+}$ influx 1 hr after RS exposure may influence the required level of depolarization for long-term maintenance of LTP (*Figure 6—figure supplement 1*).

This study demonstrated that LAGR is indispensible for stress-induced metaplasticity in the LA and for controlling fear responses to salient cues depending on the recent stress history of an animal. Individuals with lower levels of cortisol are more susceptible to developing post-traumatic stress disorder (PTSD) (*Yehuda, 2006*, *1998*), suggesting that glucocorticoid signaling dysfunction is involved in the development of PTSD. Our findings contribute to understanding how emotional memory is formed under stress conditions and provide clues for elucidating the pathophysiology and etiology of stress-related disorders, such as PTSD.

## Materials and methods

### Key resources table

| Reagent type (species) or resource | Designation | Source or reference | Identifiers |
|---|---|---|---|
| Gene (Mus musculus) | Nr3c1 | Mouse Genome Informatics http://www.informatics.jax.org/marker/MGI:95824 | MGI:95824; NCBI Gene: 14815 |
| Gene (Mus musculus) | Grp | Mouse Genome Informatics http://www.informatics.jax.org/marker/MGI:95833 | MGI:95833; NCBI Gene: 225642 |
| Strain (C57BL/6-Tg(CAG-flpe) 36Ito/ItoRbrc) | CAG-FLPe mouse | RIKEN BioResource Center | RRID:IMSR_RBRC01834 |

*Continued on next page*

Continued

| Reagent type (species) or resource | Designation | Source or reference | Identifiers |
| --- | --- | --- | --- |
| Strain (C57BL/6) | CAG-CAT-Z mouse | Proc Natl Acad Sci U S A 92: 160–164,1995 | |
| ES cell line | RENKA | Eur. J. Neurosci. 24: 2177–2190, 2006 | |
| Antibody | Rabbit anti-GR antibody | Santa Cruz Biotechnology Cat# sc-8992 | RRID:AB_2155784 |
| Antibody | Anti-NeuN antibody | Millipore Cat# MAB377 | RRID:AB_2298772 |

## Construction of *Grp-iCre* targeting vector

To construct a codon-improved Cre-recombinase (iCre)-splice gene, the intron sequence from the SV40 t-antigen gene was amplified by PCR and inserted between codons 283 and 284 of iCre. We obtained the bacterial artificial chromosome (BAC) clone containing the gastrin-releasing peptide gene (*Grp*) from the BACPAC Resources Center CHORI (Oakland, CA). Using a counter-selection BAC modification kit (Gene Bridges, Dresden, Germany), we inserted the iCre-splice gene linked to the pgk-neo cassette flanked by two FRT sites into *Grp* gene at the +1 position (+1 corresponding to A of ATG, the initiation site of translation in the *Grp* gene). The modified BAC DNA fragment inserted with the iCre-splice-pgk-neo cassette and containing the ~5.1 kbp upstream (−5079 to −1) and ~4.9 kbp downstream (+2 to +4933) sequences of *Grp* gene were cloned into pMC1DTApMA (*Kitayama et al., 2001a*) to generate the targeting vector pTVGRP-iCre.

## Construction of *Nr3c1<sup>loxP</sup>* targeting vector

A BAC clone containing the *Nr3c1* gene was obtained from BACPAC Resources Center CHORI. The 5' arm of ~5.2 kbp (−5344 to −104, +1 corresponding to the first nucleotide of the exon 3 of the *Nr3c1* gene) and the 3' arm of ~5 kbp (+104 to +5092) were subcloned into the pDONR P4-P1R and pDONER P2R-P3 vectors (Invitrogen, Carlsbad, CA), respectively, using the counter-selection BAC modification kit. A 370 bp *Nr3c1* gene fragment containing exon 3, part of intron 2, and part of intron 3 was amplified by PCR and was subcloned between two loxP sequences of a modified pDONR 221 vector containing a pgk-Neo cassette flanked by two FRT sites. To construct the targeting vector, these three plasmids were directionally subcloned into pDEST R4-R3 vector containing the diphtheria toxin gene (MC1-DTA) by MultiSite Gateway LR recombination reaction.

## ES cell culture and generation of Grp-iCre and GR<sup>loxP</sup> mice

The targeting vectors linearized with *NotI* were electroporated into the C57BL/6-derived ES cell line RENKA (*Fukaya et al., 2006*) as previously described. After the selection with G418 (150 µg/mL), the recombinant ES cell clone was identified by Southern blot analysis. The obtained recombinant ES cells were injected into 8-cell-stage embryos of ICR mice. The embryos were cultured to blastocysts and transferred to the uterus of pseudopregnant ICR mice. The resulting chimeric mice were mated with CAG-FLPe mice (provided by RIKEN BRC through the National Bio-Resource Project of the MEXT, Japan) to delete the pgk-neo gene and establish the mutant mouse line. Grp-iCre and GR<sup>loxP</sup> mice were genotyped using polymerase chain reaction (PCR) amplification of genomic DNA prepared from the mouse tail. The Grp-iCre mice were genotyped using the following primer sequences: Gic1, 5'-GTCGAGAGCTCTGAGGGTTT-3'; Gic2, 5'-GATTTCATGATCGGGACACTTACCCA-3'; Gic3, 5'-GGTACAGGAGGTAGTCCCTCACATC-3'; NeoP1, 5'-GGGGATCCACTAGCGAATTCGAAG-3'; and NeoP2, 5'-GGATCCCCTCAGAAGAACTCGTCAA-3'. The GR<sup>loxP</sup> mice were genotyped using the following primer sequences: GR3'LU, 5'-AAAAGAGCAGTGGAAGGTAGTGTG-3' and GR3'LL, 5'-TACTTCTGTCCCTTCCCAATGGAG-3'.

Animal care and experimental protocols were approved by the Animal Experiment Committee of the University of Toyama (Authorization No. 2013 MED-18 and MED-68) and carried out in accordance with the Guidelines for the Care and Use of Laboratory Animal of the University of Toyama. Three month-old male mice were used for analyses in a genotype blind manner. The mice were kept in a temperature- and humidity-controlled room under 12 hr light/dark cycle (lights on at 7:00 AM) and had ad libitum access to food and water.

## Analysis of Cre-mediated recombination efficacy

We used the mutant mice carrying both *Grp-iCre* and *CAG-CAT-Z* transgene heterozygously as the reporter mice. The mice were deeply anesthetized with pentobarbital sodium (100 mg/kg body weight, intraperitoneal injection), and then transcardially perfused with phosphate buffered saline (PBS, pH 7.4) followed by 4% paraformaldehyde (PFA) in 0.1 M phosphate buffer (PB). Brains were removed, post-fixed with the 4% PFA overnight, and dipped in 0.1 M PB containing 30% sucrose for 36 hr at 4°C. The brains were cut into 1-mm-thick coronal sections and stained for β-galactosidase (β-gal) activity for 18 hr in PBS containing 5 mM potassium hexacyanoferrate (III), 5 mM potassium hexacyanoferrate (II), 2 mM $MgCl_2$, and 1 mg/ml 5-bromo-4-chloro-3-indolyl-β-d-galactoside (*Tsujita et al., 1999*). After the staining reaction, 20-μm-thick sections were prepared from 1-mm-thick sections and mounted on glass slides. Images were obtained using a Keyence BZ-8000 microscope (Keyence, Osaka, Japan).

## Immunofluorescence staining

The brains were removed, fixed, and cryoprotected as described above and then cut into 25-μm-thick serial coronal sections using a freezing microtome. For each animal, every fourth section (bregma from −1.4 to −1.8 mm, five sections per animal) was selected on the basis of the Mouse Brain in Stereotaxic Coordinates (*Paxinos and Franklin, 2001*) and was prepared for immunofluorescence staining.

Free floating brain sections were rinsed with PBS and blocked with Protein Block Serum Free (DAKO Cytomation, Carpinteria, CA) for 10 min at room temperature. The sections were then incubated with primary antibodies (rabbit anti-GR, 1:100, Santa Cruz #H-300; mouse anti-NeuN, 1:100, Millipore #MAB377) diluted in PBS containing 1% bovine serum albumin (BSA) overnight at 4°C. After washing with PBS, the sections were incubated with Alexa-Fluor-488- and Alexa-Fluor-594-conjugated species-specific secondary antibodies (Invitrogen) for 1 hr at room temperature. Then, the sections were washed with PBS, adhered to glass slides and coverslipped. Images were taken using a confocal laser scanning microscope (Leica TCS-SP5, Leica Microsystems, Germany). Three sections per animal were used for quantification of GR-positive and NeuN-positive cells in the LA. All sections were counted by us who were blind to the genotype of the mice.

## Auditory-cued fear conditioning

Except for the optogenetic experiment, auditory-cued fear conditioning (AFC) was conducted in a small conditioning chamber (10 × 10 × 10 cm; with transparent walls and a floor made of 14 stainless steel rods) surrounded by a sound-attenuating chest (CL-M3, O'Hara and Co., Ltd, Tokyo, Japan). Mice were placed in the conditioning chamber for 60 s and then presented with a 30 s tone (65 dB, white noise) through a speaker on the ceiling of the sound-attenuating chamber. At the end of the tone presentation, mice were given a foot shock (0.5 mA, 2 s for three CS × US pairings at 1 min intervals and 0.5 mA, 1 s for six CS × US pairings at 30 s intervals). Mice were returned to their home cages 30 s after the last foot shock. The contextual and auditory-cued fear memory retention tests were conducted 24 and 48 hr after training, respectively. During the contextual memory test, mice were re-exposed to the conditioning chamber for 5 min without the tone and shock presentation. The conditioning chamber was cleaned with 70% ethanol before each trial. For the auditory-cued memory test, mice were placed in a novel chamber (10 × 10 × 10 cm, with white walls and a flat floor). Freezing was scored for 3 min before delivery of the tone and then scored for 3 min in the presence of the tone. Before each trial, the chamber was cleaned with 4% acetic acid. Freezing responses were analyzed on a Macintosh computer with Image FZC 2.22sr2 (O'Hara and Co., Japan), which is software based on the NIH Image program.

For the optogenetic experiment, AFC was conducted in a soundproof experimental room (Yamada Co., Japan). Mice were placed in the square conditioning chamber (17.5 × 30 cm, with a transparent front wall, white side and back walls, and a floor of 26 stainless steel rods) for 60 s and then received three pairings of CS (65 dB, 7 kHz, 30 s tone) and US (0.5 mA, 2 s foot shock) with 1 min inter-pairing interval. Mice were returned to their home cages 90 s after the last foot shock. Before the first auditory-cued fear memory test (Test 1), mice were anaesthetized with isoflurane and optic fibers were inserted into guide cannulas. Mice were returned to their home cages and left for at least 1 hr. Test 1 was conducted in a circular chamber (opaque black walls, diameter 21.5

cm × height 34 cm, with a flat floor) 24 hr after conditioning. Immediately after Test 1, mice received the optical LTP or LTD protocol and 24 hr later, the second auditory-cued fear memory test (Test 2) was conducted in a novel circular chamber (opaque reddish walls, diameter 23.5 cm × height 22.5 cm, with a flat floor). Freezing was scored for 3 min before delivery of the tone and then scored for 3 min in the presence of the tone.

## Open field test

The open field test was performed to measure locomotion and anxiety in a novel open field using SCANET MV-40 (MELQUEST Co., Toyama Japan). The open field consisted of a square box of 48 cm per side with infrared sensors arranged at 6 mm intervals for the automatic detection of activity. Individual mice were placed at the center of the open field at the initiation of the test session and allowed to freely explore the field for 15 min.

## Restraint stress

Mice were placed in an adequately ventilated 50 mL plastic tube (FALCON) for 20 min. They could rotate from a prone to supine position and back again but not turn head to tail.

## Corticosterone measurements

Plasma was extracted from blood samples collected from the heart. All blood samples were collected between 2 and 4 p.m. Blood samples were centrifuged at 3,000 rpm for 5 min at 4℃ and extracted plasma was stored at −30℃ until assayed for corticosterone using a corticosterone enzyme immunoassay kit (Arbor Assays, Ann Arbor, MI) in accordance with the manufacturer's instructions.

## Viral vector

The GR-expressing adeno-associated virus (AAV-GR) and green fluorescent protein-expressing AAV (AAV-GFP) plasmids consisted of the mouse synapsin I promoter followed by cDNA encoding GR or the GFP and a woodchuck hepatitis virus post-transcriptional regulatory element between the inverted terminal repeats of the AAV3 genome. Plasmid AAV-hsyn-oChIEF-tdTomato was purchased from Addgene (Cambridge, MA). Recombinant AAV vectors were produced by transient transfection of HEK293 cells with the vector plasmid, an AAV3 rep and AAV9 vp expression plasmid, and an adenoviral helper plasmid, pHelper (Agilent Technologies, Santa Clara, CA), as previously described (*Iida et al., 2013*; *Li et al., 2006*). Recombinant viruses were purified by isolation from two sequential continuous CsCl gradients, and the viral titers were determined by qRT-PCR. Before administration, AAV-GR and AAV-GFP vectors were diluted in phosphate-buffered saline to $5 \times 10^{11}$ genome copies/100 μL and $1 \times 10^{11}$ genome copies/100 μL, respectively.

## Surgery

Mice were deeply anesthetized with pentobarbital sodium (100 mg/kg body weight, intraperitoneal injection) and placed in a stereotactic frame (Narishige, Tokyo, Japan). AAV-GR or AAV-GFP (0.3 μL per hemisphere) was bilaterally injected into the LA (AP: −1.7 mm; ML: 3.3 mm; DV: −4.1 to −4.2 mm) using glass pipettes (tip diameter 50 μm) connected to a compatible syringe and motorized microinjector (Narishige IMS-10). Mice were allowed to recover in their home cages for 4 weeks before behavioral experiments.

For optogenetic experiments, AAV-hsyn-oChIEF-tdTomato was bilaterally injected into the medial geniculate nucleus (AP: −3.1 mm; ML: 1.9 mm; DV: −3.5 mm) and the auditory cortex (AP: −2.7 mm; ML: 4.4 mm; DV: −3.3 mm). A total of 0.5 μL virus per hemisphere was injected. An optic fiber cannula was implanted just above the dorsal tip of the LA (AP: −1.7 mm; ML: 3.4 mm; DV: −4.1 mm) and secured to the skull with screws and dental cement.

## In vivo LTP or LTD induction

Optical LTP was induced with 10 trains of light (each train consisted of 100 pulses of light, 5 ms each, at 100 Hz) at 90 s inter-train intervals. Optical LTD was induced with 900 pulses of light, 2 ms each, at 1 Hz. After induction, mice were immediately returned to their home cages.

## In vivo recording

Eight weeks after the injection of AAV-hsyn-oChIEF-tdTomato into the MGN and AC, mice were anesthetized with pentobarbital sodium and then mounted on a stereotaxic frame. The optic fiber was glued to the recording electrode so that the tip of the fiber was 500 μm above the tip of the electrode. The optrode was inserted into the LA and the optic fiber was connected to a 473 nm laser unit. The LTP or LTD induction protocol was identical to those used in the behavioral test. After establishing a stable baseline at the recording site for 21 min (stimulation frequency at 0.033 Hz), in vivo LTP or LTD was induced, which was followed by at least 21 min of 0.033 Hz stimulation. Data were analyzed using Clampex 10.7 software. All animals were perfused after the recordings and the position of the recording sites were verified.

## Statistical analysis

Statistical analysis was performed using student's $t$-test for the comparison of two groups and one-way ANOVA was used for analyzing experiments consisting of multiple groups. Two-way repeated measures ANOVA was used for analyzing the freezing level during training sessions. Significant ANOVA results were followed by a $post\ hoc$ Tukey–Kramer test for multiple comparisons. All values are presented as mean ± S. E. M. Values of $p < 0.05$ were considered statistically significant.

Statistical analysis was performed using Statcel2. Mice were randomly assigned to experimental groups prior to experimentation. No statistical methods were used to pre-determine sample sizes but our sample sizes are similar to those generally employed in the field. Animals with viral injections off target or damage during the experiment were excluded from the statistical analysis.

## Acknowledgements

We thank Dr. R Sprengel for providing the plasmid containing iCre and Dr. J Miyazaki for providing the CAG-CAT-Z mice. CAG-FLPe mice were provided by the RIKEN BRC through the National Bio-Resource Project of the MEXT, Japan. We thank M Ito and N Takiko (Jichi Medical University) for their help with production of the AAV vectors. This work was supported by Grant-in-Aid for Scientific Research on Innovative Areas Nos. 221S0003, 15H01279, 25116508, and JP25115002, Grant-in-Aid for Young Scientists (B) Nos.25830005 and 26293213, a Grant-in-Aid for Scientific Research (C) No. 15K06705 from the Ministry of Education, Culture, Sports, Science and Technology of Japan, a grant from the Japan Agency for Medical Research and Development (AMED), and a grant from the Core Research for Evolutional Science and Technology (CREST) program (JPMJCR13W1) of the Japan Science and Technology Agency (JST).

## Additional information

### Competing interests

Shin-ichi Muramatsu: SM owns equity in a gene therapy company (Gene Therapy Research Institution) that commercializes the use of AAV vectors for gene therapy applications. To the extent that the work in this manuscript increases the value of these commercial holdings, SM has a conflict of interest. The other authors declare that no competing interests exist.

### Funding

| Funder | Grant reference number | Author |
| --- | --- | --- |
| Ministry of Education, Culture, Sports, Science, and Technology | 26293213 | Ran Inoue |
| Ministry of Education, Culture, Sports, Science and Technology | 15K06705 | Ran Inoue |
| Ministry of Education, Culture, Sports, Science, and Technology | JP25115002 | Kaoru Inokuchi |

| Core Research for Evolutional Science and Technology | JPMJCR13W1 | Kaoru Inokuchi |
| Ministry of Education, Culture, Sports, Science, and Technology | 221S0003 | Hisashi Mori |
| Ministry of Education, Culture, Sports, Science and Technology | 15H01279 | Hisashi Mori |
| Ministry of Education, Culture, Sports, Science and Technology | 25116508 | Hisashi Mori |

The funders had no role in study design, data collection and interpretation, or the decision to submit the work for publication.

## Author contributions

Ran Inoue, Conceptualization, Resources, Data curation, Software, Formal analysis, Supervision, Funding acquisition, Investigation, Methodology, Writing—original draft, Project administration, Writing—review and editing; Kareem Abdou, Data curation, Investigation; Ayumi Hayashi-Tanaka, Shin-ichi Muramatsu, Kaori Mino, Investigation; Kaoru Inokuchi, Funding acquisition, Methodology; Hisashi Mori, Conceptualization, Supervision, Funding acquisition, Investigation, Project administration, Writing—review and editing

## Author ORCIDs

Ran Inoue (ID) http://orcid.org/0000-0003-0700-7734
Shin-ichi Muramatsu (ID) https://orcid.org/0000-0002-3185-7790
Kaoru Inokuchi (ID) https://orcid.org/0000-0002-5393-3133
Hisashi Mori (ID) http://orcid.org/0000-0001-9743-2456

## Ethics

Animal experimentation: Animal care and experimental protocols were approved by the Animal Experiment Committee of the University of Toyama (Authorization No. 2013 MED-18 and MED-68) and carried out in accordance with the Guidelines for the Care and Use of Laboratory Animal of the University of Toyama.

## Decision letter and Author response

Decision letter https://doi.org/10.7554/eLife.34135.022
Author response https://doi.org/10.7554/eLife.34135.023

# Additional files

## Supplementary files

• Transparent reporting form
DOI: https://doi.org/10.7554/eLife.34135.020

## Data availability

All data generated or analysed during this study are included in the manuscript and supporting files. Source data files have been provided for Figures 2 -6.

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
