## [Decision Letter]

Thank you for submitting your article "Glucocorticoid receptor-mediated amygdalar metaplasticity underlies adaptive modulation of fear memory by stress" for consideration by *eLife*. Your article has been reviewed by 3 peer reviewers, including a Reviewing Editor, and the evaluation has been overseen by a Senior Editor. The following individual involved in review of your submission has agreed to reveal his identity: Sumantra Chattarji (Reviewer #2).

The reviewers have discussed the reviews with one another and the Reviewing Editor has drafted this decision to help you prepare a revised submission.

Summary:

The reviewers agreed that the work is very interesting, and that the investigation into the interactions between the glucocorticoid receptor in the lateral amygdala and fear conditioning are novel and timely. The reviewers raise a number of concerns that must be adequately addressed before the paper can be accepted, and some of the required revisions will likely require further experimentation within the framework of the presented studies and techniques.

Essential revisions:

1) Figure 6: The figure legend states that "Data are obtained from four mice (two floxed GR and two LAGRKO mice)." In the LTP/LTD graphs (Figure 6F, G) why are floxed GR and LAGRKO mice data combined? It is critical to show that LTP and LTD can be elicited in both types of mice. Importantly, representative sample traces of fEPSPs (Figure 6F-G) should be shown.

2) The authors are not specific about what they mean by suppression of LTP. Is the suppression at the level of the induction or expression? The LTP experiment that they did does not differentiate between these two cases. How does the stress cause a deficit? Is this, as proposed by Karst et al., the result of GR-dependent reduction in the excitability of the LA neurons, hence the threshold for potentiation? If so, is this represented in the electrophysiological data? That is, do they see a difference in LTP-induced synaptic potentiation between stressed control vs. stressed LAGRKO? Please discuss.

3) Add discussion about what the model is; answer the following questions even if it's speculation:

- In the fourth paragraph of the Discussion, the authors state that activation of LAGR by stress suppresses fear conditioning-induced LTP at auditory inputs to the LA and that this is indicated by the reduction of freezing in restraint stress exposed LAGRKO mice after delivery of an LTD protocol. Does this imply that the "stress exposure suppression of fear-conditioned LTP" is through a synaptic mechanism similar to that elicited by optical LTD?

I don't think they can conclude that the "stress exposure suppression of fear-conditioned LTP" is through a synaptic mechanism similar to that elicited by optical LTD?

4) The absence of any data on corticosterone levels elicited by restraint stress is surprising in light of the central role this form of stress plays in the experimental findings reported in the manuscript. Further, what are the levels of corticosterone in animals that are first subjected to restrain stress followed by fear conditioning?

5) A missing piece of the puzzle is that the authors have not monitored online how in vivo correlates of synaptic strength (e.g. slope of fEPSPs evoked by, say, electrical activation of thalamic inputs from the MGN) change as a result of fear conditioning and its modulation by restraint stress and other genetic manipulations used here. Do the authors have any data to address this point? While not required for revision, this would be of interest if the data exist.

6) More explanation of the overall model is required, in particular explaining the interactions between synaptic plasticity (optogenetic or natural), fear conditioning, and GR, in the context of restraint stress as well as control conditions. Why does the optical LTP protocol not elicit an increase in freezing in the LAGRKO mice (Figure 6D) as seen with the Floxed-GR mice (Figure 6C)? Since these animals are capable of forming fear memories, the LA neurons should have the capacity to undergo synaptic plasticity. On the other hand, the authors have shown that these same LAGRKO mice have attenuated fear responses after an optical LTD protocol. The prior restraint stress suppresses the induction of synaptic potentiation during fear conditioning. The authors are not specific about what they mean by suppression of LTP. Is the suppression at the level of the induction or expression? The LTP experiment that they did, does not differentiate between these two cases. Does this imply that optical LTP itself is dependent on GRs in the LA? Is GR activation not required for LTP or LTD induction in LA?; the authors should at the least discuss where they believe GR activation occurs in the chain of learning events in the context of restraint stress. A conclusion diagram could be very helpful.

[Editors' note: further revisions were requested prior to acceptance, as described below.]

Thank you for resubmitting your work entitled "Glucocorticoid receptor-mediated amygdalar metaplasticity underlies adaptive modulation of fear memory by stress" for further consideration at *eLife*. Your revised article has been favorably evaluated by a Senior Editor and a Reviewing Editor.

The manuscript has been improved but there are some remaining issues that need to be addressed before acceptance, as outlined below:

The authors have responded to nearly all of the reviewers' points adequately. However, their response to the first critique is not acceptable. Here is the critique again:

1) Figure 6: The figure legend states that "Data are obtained from four mice (two floxed GR and two LAGRKO mice)." In the LTP/LTD graphs (Figure 6F, G) why are floxed GR and LAGRKO mice data combined? It is critical to show that LTP and LTD can be elicited in both types of mice.

The authors responded that they convinced themselves that the relevant manipulations trigger LTP or LTD in each genotype is not sufficient. The experiments in each genotype must be done separately and statistically validated (showing that statistically signifcant LTP is elicited, e.g.) – not included together. The experimental quality of the LTP experiments in particular is poor. Baselines must be stable for at minimum 10 minutes, not slanting up or jumping upward just prior to LTP induction as is apparent in 2 of the 4 experiments. Please repeat these experiments and demonstrate that the optical protocols in slices are sufficient to induce LTP and LTD in both floxed GR and LAGRKO mice. Without these data, the manuscript is lacking an important piece of information necessary to interpret the results.

---

## [Author Response]

1) Figure 6: The figure legend states that "Data are obtained from four mice (two floxed GR and two LAGRKO mice)." In the LTP/LTD graphs (Figure 6F, G) why are floxed GR and LAGRKO mice data combined? It is critical to show that LTP and LTD can be elicited in both types of mice. Importantly, representative sample traces of fEPSPs (Figure 6F-G) should be shown.

The purpose of this experiment is to confirm whether our optical LTP and LTD protocols produce expected synaptic changes in vivo. Our optical LTP and LTD protocols successfully induced LTP and LTD in all four naïve mice examined (two floxed GR and two LAGRKO mice), respectively. The data for each individual mouse are shown in Figure 6—figure supplement 1. Because there was no evident difference in LTP and LTD induction between floxed GR and LAGRKO mice, we pooled the data from both floxed GR and LAGRKO mice.

According to your suggestion, we added representative sample traces of fEPSPs in Figures 6F and G.

2) The authors are not specific about what they mean by suppression of LTP. Is the suppression at the level of the induction or expression? The LTP experiment that they did does not differentiate between these two cases. How does the stress cause a deficit? Is this, as proposed by Karst et al., the result of GR-dependent reduction in the excitability of the LA neurons, hence the threshold for potentiation? If so, is this represented in the electrophysiological data? That is, do they see a difference in LTP-induced synaptic potentiation between stressed control vs. stressed LAGRKO? Please discuss.

In this study, we did not conduct in vivo electrophysiological analysis using stressed mice. However, in a previous electrophysiological study, brain slices containing the BLA region were analyzed to understand how restraint stress (RS) affects synaptic plasticity (Sarabdjitsinh and Joels, 2014). This report demonstrated that the persistence of LTP induced in the presence of corticosterone is sensitive to a previous RS exposure (Sarabdjitsinh and Joels, 2014). Therefore, in our study, we postulated that the persistence of LTP in fear-conditioned floxed GR mice is suppressed by a previous RS exposure, thereby leading to suppression of auditory-cued fear memory. This was indirectly verified by the fact that suppressed auditory-cued fear memory in RS-exposed floxed GR mice was rescued by optogenetic LTP induction in the LA. Accordingly, such suppression of auditory-cued fear memory was not observed in RS-exposed LAGRKO mice, suggesting that the persistence of LTP was not altered in RS-exposed LAGRKO mice. Thus, optogenetic delivery of LTP to the LA of RS-exposed LAGRKO mice exerted no influence on auditory-cued fear memory.

Two studies have contributed to our understanding of the GR-dependent mechanisms involved in stress-related modulation of synaptic plasticity in the LA. Di et al. demonstrated that a short application of corticosterone activates G protein-coupled postsynaptic membrane glucocorticoid receptors (mGRs), leading to synthesis and release of an endocannabinoid that binds to presynaptic CB1 endocannabinoid receptor to suppress glutamate release (Di et al., 2003). Thus, postsynaptic mGRs activation induced by a recent RS exposure may result in reduction of glutamate release during subsequent fear conditioning. In another study, Sato et al. demonstrated that membrane-impermeable corticosterone treatment decreases *N*-methyl-D-aspartate (NMDA)-induced Ca^2+^ elevation, suggesting that activation of mGRs may inhibit Ca^2+^ influx via the NMDA receptor (Sato et al., 2004). Therefore, the reduced presynaptic glutamate release and inhibition of postsynaptic Ca^2+^ influx 1 h after RS exposure may influence the required level of depolarization for long-term maintenance of LTP. See Discussion, third paragraph. A diagram of a summary of these events was added in Figure 6—figure supplement 2.

3) Add discussion about what the model is; answer the following questions even if it's speculation:- In the fourth paragraph of the Discussion, the authors state that activation of LAGR by stress suppresses fear conditioning-induced LTP at auditory inputs to the LA and that this is indicated by the reduction of freezing in restraint stress exposed LAGRKO mice after delivery of an LTD protocol. Does this imply that the "stress exposure suppression of fear-conditioned LTP" is through a synaptic mechanism similar to that elicited by optical LTD?I don't think they can conclude that the "stress exposure suppression of fear-conditioned LTP" is through a synaptic mechanism similar to that elicited by optical LTD?

As we have addressed in our response to comment 2, activation of LAGR by RS may suppress the persistence of LTP in fear-conditioned mice, and this speculation is indirectly confirmed by our experimental data showing that optogenetic delivery of an LTP protocol increased fear responses in RS-exposed floxed GR mice. In contrast, optogenetic delivery of an LTP protocol did not alter the freezing level in RS-exposed LAGRKO mice, further suggesting that LAGR is required for the mechanisms responsible for stress-induced suppression of LTP. The purpose of using the LTD protocol is to verify whether the conditioned fear responses in RS-exposed LAGRKO mice is altered by reducing the synaptic strength. In the present study, we did not conclude that a synaptic mechanism similar to that elicited by an optical LTD is responsible for stress-induced suppression of fear-conditioned LTP.

4) The absence of any data on corticosterone levels elicited by restraint stress is surprising in light of the central role this form of stress plays in the experimental findings reported in the manuscript. Further, what are the levels of corticosterone in animals that are first subjected to restrain stress followed by fear conditioning?

We measured plasma corticosterone levels 1 h after RS exposure alone and 90 min after RS exposure followed by fear conditioning. There was no significant difference in plasma corticosterone levels between floxed GR and LAGRKO mice at the two time points. See Results, subsection “Disruption of LAGR impairs adaptive modulation of conditioned fear in response to prior restraint stress exposure” and Figure 5—figure supplement 1.

5) A missing piece of the puzzle is that the authors have not monitored online how in vivo correlates of synaptic strength (e.g. slope of fEPSPs evoked by, say, electrical activation of thalamic inputs from the MGN) change as a result of fear conditioning and its modulation by restraint stress and other genetic manipulations used here. Do the authors have any data to address this point? While not required for revision, this would be of interest if the data exist.

Owing to technical limitations, we did not monitor changes in synaptic strength online as a result of fear conditioning and its modulation by restraint stress in floxed GR and LAGRKO mice. We hope to perform such an experiment in the near future.

6) More explanation of the overall model is required, in particular explaining the interactions between synaptic plasticity (optogenetic or natural), fear conditioning, and GR, in the context of restraint stress as well as control conditions. Why does the optical LTP protocol not elicit an increase in freezing in the LAGRKO mice (Figure 6D) as seen with the Floxed-GR mice (Figure 6C)? Since these animals are capable of forming fear memories, the LA neurons should have the capacity to undergo synaptic plasticity. On the other hand, the authors have shown that these same LAGRKO mice have attenuated fear responses after an optical LTD protocol. The prior restraint stress suppresses the induction of synaptic potentiation during fear conditioning. The authors are not specific about what they mean by suppression of LTP. Is the suppression at the level of the induction or expression? The LTP experiment that they did, does not differentiate between these two cases. Does this imply that optical LTP itself is dependent on GRs in the LA? Is GR activation not required for LTP or LTD induction in LA?; the authors should at the least discuss where they believe GR activation occurs in the chain of learning events in the context of restraint stress. A conclusion diagram could be very helpful.

Please refer to our response to comment 2. A conclusion diagram is shown in Figure 6—figure supplement 2.

[Editors' note: further revisions were requested prior to acceptance, as described below.]

The manuscript has been improved but there are some remaining issues that need to be addressed before acceptance, as outlined below:The authors have responded to nearly all of the reviewers' points adequately. However, their response to the first critique is not acceptable. Here is the critique again:1) Figure 6: The figure legend states that "Data are obtained from four mice (two floxed GR and two LAGRKO mice)." In the LTP/LTD graphs (Figure 6F, G) why are floxed GR and LAGRKO mice data combined? It is critical to show that LTP and LTD can be elicited in both types of mice.The authors responded that they convinced themselves that the relevant manipulations trigger LTP or LTD in each genotype is not sufficient. The experiments in each genotype must be done separately and statistically validated (showing that statistically signifcant LTP is elicited, e.g.) – not included together. The experimental quality of the LTP experiments in particular is poor. Baselines must be stable for at minimum 10 minutes, not slanting up or jumping upward just prior to LTP induction as is apparent in 2 of the 4 experiments. Please repeat these experiments and demonstrate that the optical protocols in slices are sufficient to induce LTP and LTD in both floxed GR and LAGRKO mice. Without these data, the manuscript is lacking an important piece of information necessary to interpret the results.

We performed additional electrophysiological experiments to show that the optical protocols are sufficient to induce LTP and LTD in both floxed GR (n = 4) and LAGRKO GR (n = 4) mice, respectively. See Results, subsection “Optogenetic manipulation of metaplasticity in the LA alters the behavioral effect of 186 prior RS exposure on auditory-cued fear memory”, last paragraph; Figure 6 and figure legend. The original data for Figure 6F, G are provided in Figure 6—source data 2.